# Stain Standardization Capsule: A pre-processing module for histopathological image analysis

Yushan Zheng[1,2], Zhiguo Jiang[2,1], Haopeng Zhang[2,1],
Jun Shi[3], and Fengying Xie[2,1]

[1] Beijing Advanced Innovation Center for Biomedical Engineering, Beihang University, Beijing, 100191, China {yszheng, jiangzg}@buaa.edu.cn
[2] Image Processing Center, School of Astronautics, Beihang University, Beijing, 102206, China
[3] School of Software, Hefei University of Technology, Hefei 230601, China

**Abstract.** Color consistency is crucial to developing robust deep learning methods for histopathological image analysis. With the increasing application of digital histopathological images, the deep learning methods are likely developed based on the data from multiple medical centers. This requirement makes it a challenge task to normalize the color variance of histopathological images from different medical centers. In this paper, we proposed a novel color standardization module named stain standardization capsule (SSC) based on the paradigm of capsule network and the corresponding dynamic routing algorithm. The proposed module can learn and generate uniform stain separation outputs for histopathological images in various color appearance without the reference to manually selected template images. The SSC module is light and can be trained end-to-end with the application-driven CNN model. The proposed method was validated on two public datasets and compared with the state-of-the-art methods. The experimental results have demonstrated that the SSC module is effective in color normalization for histopathological images and achieves the best performance in the compared methods.

**Keywords:** Stain standardization · Histopathological image analysis · Digital pathology · Capsule network.

## 1  Introduction

Based on the widespread application of digital pathology (DP) in cancer research and clinical diagnosis, an increasing number of methods for histopathological image analysis (HIA) have been proposed. In practice, color appearance of digital whole slide images (WSI) varies due to the diversity in the section fabrication and digitization, which makes it a challenge task to establish robust analysis frameworks for digital histopathological images from different medical centers. Generally, stain standardization (or normalization) is the main approach to solve the problem of stain color variances.

The early studies utilized the color style transformation methods in natural scene image processing and concentrated on matching the color of one histopathological image patch to another one [7, 5]. With the development of whole slide imaging techniques, the requirement of computer-aided diagnosis has changed from histopathological image patches to WSIs. Simultaneously, the stain transformation algorithms for WSIs are developed [1, 15, 16]. The color transform parameters were estimated or optimized with abundant pixels sampled from the entire WSI. The robustness of stain normalization has been greatly improved compared to the patch-based transformation methods [7, 5, 3]. However, the dependence of abundant pixels and whole slide images meanwhile narrowed the scope of application.

Recently, the data-driven deep-learning methods, especially the convolutional neural networks (CNNs), have become the major basics of emerging HIA researches. Correspondingly, the requirement of data standardization is further promoted to adapt multiple stain domains from different datasets provided by different medical centers. One popular scheme to solve the dataset-wise color variance is color domain transfer, where the algorithms based on generative adversarial networks (GANs) are widely studied [14, 10]. Instead of estimating transform parameters between image pairs or WSI pairs, these methods established a GAN structure to learn the data adaption principle between the training dataset and application (testing) dataset. The performance of stain standardization has proven very promising. Nevertheless, the present GAN-based methods require to know the full data distribution of the application dataset and the transform model is required to be trained in pairs if there are more than two medical centers providing the data. Another scheme to solve the problem is color augmentation. Tellez et al. [11] proposed a stain augmentation strategy based on CD theory to simulate different staining situations, which has proven effective in improving the generalization ability of the CNN model for stain variance. However, the stain information is extracted using fixed model parameters that estimated under ideal dyeing case. When facing the samples in non-ideal situation, the augmented samples would be out of the distribution of real cases. Another study [12] constructed an U-Net model to learn an uniform color style from images with random color biases. The trained network is powerful in the color normalization of unseen histological images. While, the network contains millions of parameters, which makes it less efficient in computation.

Facing the current issues in the multiple stain domain standardization, we proposed a novel stain standardization module for CNN-based histopathological image analysis, which is named as stain standardization capsule (SSC). The basic theory of SSC is the stain separation in optical density space [8]. The structure of the module is modified from the *Capsule Network* [9] and the stain standardization is realized referring to the dynamic routing (DR) operations in the capsule network (as shown in Fig. 1). The contribution of this paper and novelty to the existing methods can be summarized as follows.

1) We brings the insight of dynamic routing into histopathological image standardization. Beyond optimizing the normalization parameters for specific

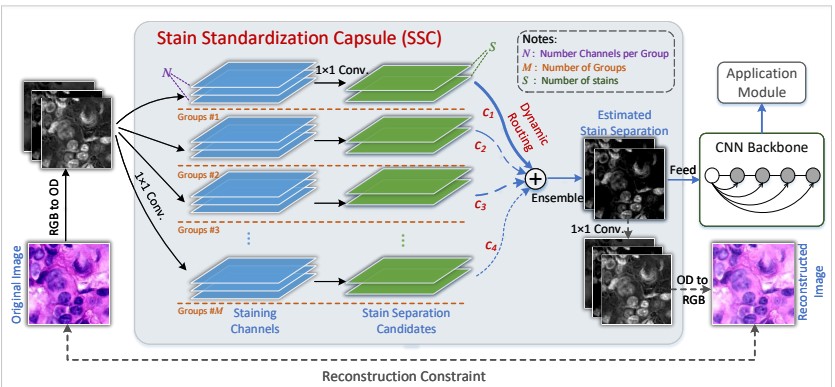

**Fig. 1.** Structure of the proposed SSC module, where the input RGB-format images are first converted to the optical density space, then projected into $M$ groups of $S$ stain channels via linear transformations, and finally assembled to obtain the stain separation results via the designed sparsity routing algorithm.

image (or WSI) [5, 1, 13, 16] or estimating the color transfer model depending on plenty of samples from the application dataset [14, 10], the proposed SSC module automatically summarizes a set of candidate ways to stain separation based on the training data that involves various color appearance. In the application stage, the stain standardization is achieved by optimizing the forward route within the pre-trained candidate ways via the designed sparsity routing process. It prevents the standardization results from serious artifacts and even failures.

2) The SSC module is much lighter (containing only tens of parameters) than CNN-based methods [12, 14, 10] and can be trained end-to-end with specific HIA tasks. Furthermore, the module does not need manually selected template images, which determines the SSC module is easy-to-use in both the development and deployment of HIA applications.

3) The proposed method is evaluated on two public datasets and compared with the sate-of-the-art methods. The experimental results have demonstrated the effectiveness and advantages in developing HIA applications.

## 2   Method

The approach of the proposed SSC module to stain standardization is achieved by generating uniform stain separation tensors for images in various color appearance. A set of stain separation candidates are first constructed and the separation result is obtained by a weighted sum of these candidates.

### 2.1   Stain separation candidates

Color deconvolution (CD) [8] is a popular stain separation method for digital slides where the staining dyes obey Beer-Lambert law. CD is utilized as the

basic theory of popular stain standardization methods [5, 13, 16]. Referring to [8], independent stain components can be extracted through linear transformation in the optical density (OD) space. Hence, we constructed a CNN structure with linear projection operations to learn possible stain extraction principles in OD-space based on all the training images. Then, we assigned the stain extraction layers into $M$ groups with the same structures, generating $M$ stain separation candidates. The detail of the SSC structure is illustrated in Fig. 1.

Letting $\mathbf{o} \in \mathbb{R}^{m \times n \times 3}$ denote the optical density of an image in size of $m \times n$ pixels [4], the grouped linear projections can be represented as

$$\mathbf{u}_i = Conv(\mathbf{o}, \mathbf{W}_i^{(1)}) \in \mathbb{R}^{m \times n \times N},$$

$$\hat{\mathbf{u}}_i = Conv(\mathbf{u}_i, \mathbf{W}_i^{(2)}) \in \mathbb{R}^{m \times n \times S}, i = 1, 2, ..., M,$$

where $Conv$ represents a convolution operation followed by a leakly-relu activation, $\mathbf{W}_i^{(1)}$ and $\mathbf{W}_i^{(2)}$ are the convolutional weights, and $M$, $N$ and $S$ denotes the number of groups, the number of channels in the first convolution and the number of stains involved in the images, respectively.

## 2.2   Sparsity routing

*Capsule Network* is a new paradigm of artificial neural networks proposed by Hinton et al. [9], in which the input of the neurons are defined as a set of vectors, rather than scalars that defined in traditional neural networks. The set of vectors are assembled by a weighted sum operation and then activated. And the weights of the input vectors for ensemble are decided by the dynamic routing (DR) algorithm.

Motivated by the insight of capsule network, we propose assembling the stain candidates $\{\hat{\mathbf{u}}_i | i = 1, ..., M\}$ through DR. The aim of the routing is to find the most appreciate stain separation result for each specific image from the $M$ candidates in the forward way of the network.

Generally, a good stain separation should exclusively assign the value of a pixel to one stain channel, i.e. the separated result is desired to be pixel-wise sparse [5, 1, 16]. Therefore, we designed a novel *Sparsity Routing* (SR) algorithm by modifying the agreement scoring in DR. The pseudo-code of SR is given in Algorithm 1. The score of the pixel-wise sparsity is calculated based on the sparseness measure defined in [4]:

$$\eta_p(\mathbf{x}) = \frac{1}{mn} \sum_i \sum_j \frac{\sqrt{S} - \sum_k |x_{ijk} + \epsilon| / \sqrt{\sum_k (x_{ijk} + \epsilon)^2}}{\sqrt{S} - 1},$$

where $\mathbf{x} \in \mathbb{R}^{m \times n \times S}$ denotes the tensor to score. To avoid all the image data being assigned to a single stain channel, a channel-wise sparseness is additionally

---

[4] $\mathbf{o} = -\log(\mathbf{I} + \epsilon)/I_{max}$, where $\mathbf{I}$ represents a RGB-format image, $I_{max}$ is the upper intensity for the digitization and $\epsilon$ is a small scalar to protect the log operation.

defined:

$$\eta_c(\mathbf{x}) = \frac{1}{S}\sum_k \frac{\sqrt{mn} - \sum_i \sum_j |x_{ijk} + \epsilon| / \sqrt{\sum_i \sum_j (x_{ijk} + \epsilon)^2}}{\sqrt{mn} - 1}.$$

Then, the sparsity score is formulated as $\eta(\mathbf{x}) = \eta_p(\mathbf{x}) + \eta_c(\mathbf{x})$ and referred as $SparseScore(\mathbf{x})$ in Algorithm 1. After SR, the output of SSC is calculated by equation

$$\mathbf{s} = \sum_{i=1}^{M} c_i \cdot \hat{\mathbf{u}}_i.$$

The SR process allows SSC generating refined stain separation results by tuning the weights $\{c_i\}$ and then allows the following CNNs concentrate on the structural variances of tissue images.

---

**Data:** $\{\hat{\mathbf{u}}_i | i = 1, ..., M\} \leftarrow$ The grouped outputs of the candidate layer;
$R \leftarrow$ The number of routings;
**SparseRouting**$(\{\hat{\mathbf{u}}_i\}, R)$:
for all the group $i$ in the candidate layer: $b_i \leftarrow 0$, $c_i \leftarrow 1/M$;
**for** $r = 1$ *to* $R$ **do**
    $\hat{\mathbf{s}} \leftarrow \sum_i c_i \cdot \hat{\mathbf{u}}_i$;
    for all the group $i$ in the candidate layer: $b_i \leftarrow b_i + SparseScore(\hat{\mathbf{u}}_i + \hat{\mathbf{s}})$;
    for all the group $i$ in the candidate layer: $c_i \leftarrow \exp(b_i)/\sum_i \exp(b_i)$;
**end**
return $\{c_i\}_1^M$;

**Algorithm 1:** The algorithm of sparsity routing.

## 2.3   Training and Application of SSC

The SSC module is essentially a convolutional neural network. Therefore, it can be directly equipped to an application-driven CNN and trained end-to-end along with the target of the CNN. The assembled stain separation result $\mathbf{s}$ is the output of the SSC module and meanwhile the input of the following CNN. To ensure $\mathbf{s}$ preserves the structural information of the histopathological image, a reconstruction layer is appended to the end of SSC and a mean square error (MSE) loss is considered between the original image and the reconstructed results. The MSE loss is merged to the loss of the following CNN in the training stage. Note that the SR only processes in the forward stage and the scalars $c_i$ are constant in the backward stage [9].

## 3   Experiment

### 3.1   Experimental settings

The proposed SSC module was validated on *Camelyon16*[5] and *ACDC-LungHP*[6] datasets [2, 6] via histopathological image classification tasks. Regions with cancer in the WSIs are annotated by pathologists. Image patches in size of $224 \times 224$ were randomly sampled from the WSIs. Patches containing above 75% cancerous pixels according to the annotation were labeled as positive and containing none cancerous pixels were labeled as negative. The other patches were not used in the experiments.

The DenseNet-121 CNN structure with softmax output was employed for classification. The sensitivity, specificity, accuracy and the area under ROC carve were used for evaluation metrics. 20% samples in the training set were spared for validation and the remainders were used to train the model. The hyper-parameters $M, N, R$ in SSC were tuned in the training set and determined according to the classification error of the validation samples. Specifically, $(M, N, R)$ is determined as $(5, 3, 3)$ for Camelyon16 and $(4, 3, 4)$ for ACDC-LungHP. $S$ is set to 2 because the images are all from H&E-stained histology.

### 3.2   Results and discussion

The classification performance in the Camelyon16 testing set are presented in Table 1, where three state-of-the-art methods [11, 12, 16] are compared[7]. Table 1 also provides a summary on the dependence and the property of each compared method. Overall, our SSC module is the most effective in improving the classification performance. The performance of data standardization appeared to be less effective in ACDC-lungHP dataset than in Camelyon16 dataset since the color consistency in the former dataset is relatively better than the latter.

*Stain augmentation* [11] utilized the prior knowledge of slide staining to augment the color allocation of training images. Therefore, the classification network using *Stain augmentation* [11] achieved better classification metrics than that using a common *Color augmentation* (including random illumination, saturation, hue and contrast transfers in the experiment) method. However, the method would generate images with unreasonable color styles. These samples would perform as noises in the CNN training and reduce the classification accuracy when the color distribution is originally consistent (referring to results in ACDC-lungHP dataset). ACD [16] and *CNN-norm* [12] have achieved competitive results. Nevertheless, ACD requires individually estimating standardization parameters for specific testing image and relies on the context information of the corresponding WSI. *CNN-norm* learned a general principle for images in

---

[5] https://camelyon16.grand-challenge.org/

[6] https://acdc-lunghp.grand-challenge.org/. Since the annotations of testing part of the data set are not yet accessible, only the 150 training WSIs of the data were used in this paper.

[7] the compared methods have been introduced in brief in section 1.

**Table 1.** Standardization performance for histopathological image patch classification, where the model properties, including the number of model parameters ($n_{param}$), whether to rely on manually selected templates (T.) and whether the model parameters for testing images require to be estimated (E.) are compared.

| Methods | Camelyon16 | | | | ACDC-LungHP | | | | Dependence | |
|---|---|---|---|---|---|---|---|---|---|---|
| | Sen | Spe. | Acc | AUC | Sen | Spe. | Acc | AUC | $n_{param}$ | T./E. |
| Origin | 0.851 | 0.969 | 0.910 | 0.957 | 0.822 | 0.779 | 0.801 | 0.882 | – | – |
| Color Aug | 0.868 | 0.950 | 0.909 | 0.958 | 0.836 | 0.760 | 0.798 | 0.881 | None | No/No |
| Stain Aug[11] | 0.875 | 0.946 | 0.911 | 0.967 | 0.819 | 0.778 | 0.799 | 0.882 | None | No/No |
| ACD[16] | 0.892 | 0.944 | 0.918 | 0.968 | 0.836 | 0.776 | **0.805** | 0.886 | $< 10^1$ | Yes/Yes |
| CNN-norm[12] | 0.875 | **0.970** | 0.922 | 0.971 | 0.821 | **0.788** | 0.804 | 0.886 | $> 10^7$ | Yes/No |
| SSC (Ours) | **0.894** | 0.966 | **0.930** | **0.975** | **0.840** | 0.778 | **0.805** | **0.887** | $< 10^2$ | No/No |

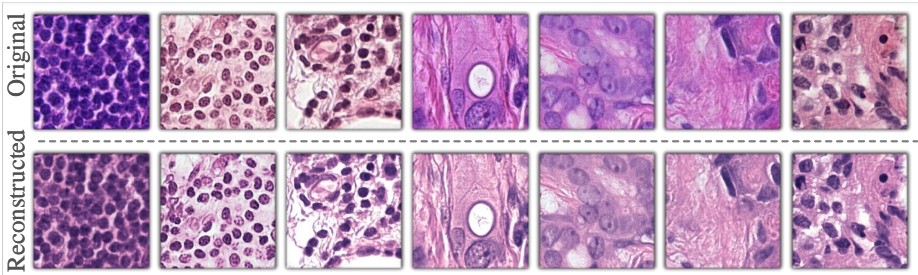

**Fig. 2.** Joint display of the original images and the reconstructed images.

different color styles with millions of model parameters ($> 10^7$). The computation amount of *CNN-norm* is comparable or even more than the following HIA application.

In comparison, our SSC module involves only tens of model parameters, does not rely on contextual information out the scope of the testing image, has no additional parameter estimation process in the prediction stage, and can be trained in end-to-end fashion. These properties determine the SSC module is more efficient and convenient than the present methods in both the training and deployment for HIA applications. Figure 2 illustrated original images and the corresponding reconstructed images from Camelyon16 dataset. Without any template images, SSC appears to have learned a "Mean" stain style in the reconstruction layer for images in diverse color appearance. It indicates an uniform representation of the SSC output layer, which has allowed the following CNN concentrating on structural discrimination in histopathological images and thus has improved the performance of the HIA application.

## 4   Conclusion

In this paper, we proposed a novel stain standardization module named stain standardization capsule for histopathological image analyis based on the op-

tical properties of tissue section staining and the insight of dynamic routing from capsule network. The proposed module is implemented in the domain of convolutional neural network and therefore can be directly equipped to CNN-based HIA application. The proposed method was evaluated with application of histopathological image classification on two public datasets. The results have demonstrated the effectiveness and robustness of the proposed methods.

## Acknowledgment

This work was supported by the National Natural Science Foundation of China (No. 61901018, 61771031 and 61906058), China Postdoctoral Science Foundation (No. 2019M650446) and Motic-BUAA Image Technology Research Center.

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
