# OpenReview forum: "Stain Standardization Capsule: A pre-processing module for histopathological image analysis"
_MICCAI.org/2019/Workshop/COMPAY — COMPAY 2019_

### Official Review · AnonReviewer2 · 2019-08-13
**Interesting methodology.**

**Rating:** 7
**Confidence:** 4

**Review:**

Quality: Well written and well-structured

Clarity: Generally clear. The experimental setting is not clear: what is the classification task for each dataset? The number of samples? How do the training and validation perform? It's not clear that the validation between different algorithm is fair. If the proposed method been trained end-to-end for the classification task, have other been trained end-to-end?

Originality: The application of capsule net to stain standardization is novel and the use of sparsity constrains in the dynamic routing is very original.

Significant of this work: High because it discards the need for the target colour distribution.

Pros:
- A very attractive approach to stain standardization: 1) less no. parameters, 2) no need of a template, and 3) can be trained end-to-end to a specific task.

Cons:
- The details for the experimental setting need to more clarity. See above.

---

### Official Review · AnonReviewer4 · 2019-08-15

**Rating:** 6
**Confidence:** 3

**Review:**

This paper is focusing on an important concept in medical imaging: image standardization/normalization which is important for suppressing difference caused by data preparation or imaging procedure and let the deep learning pipeline to focus on application purposes.

Besides evaluating classification benefits, could authors directly evaluate the quality for standardization?
In the step of SSC, is it possible to skip the step of generating Ui ?
For people who are not familiar with DR , please elaborate this.
It is not clear the motivation of bi in Algorithm 1, please explain.

---

### Official Review · AnonReviewer1 · 2019-08-19
**Stain Standardization Capsule: A pre-processing module for histopathological image analysis**

**Rating:** 7
**Confidence:** 3

**Review:**

Clarity:  The authors can provide more clarity on how the training is done.

Summary: Color standardization module, based on capsule network and dynamic routing algorithm is proposed in the context of follow-up CNN for tumor cell-classification application. The combined network is trained together and evaluated, using two public datasets of Camelyon-16 and ACDC-LungHP, and compared with alternative methods, including the one without any color standardization.

Results: SSC model is shown to be relatively slightly better in regards to classification metrics with fewer number of parameters and not requiring additional manual input.  (From the reported results, I notice that the results from different methods are quite similar and the method with any color standardization is not all that bad - which leads me to guess that the classification CNN network has the implicit capacity to model and accommodate the stain color variability. So, that raises the broader question - is color normalization required for CNN based histopathological image analysis).

Originality:  SSC module is an interesting and attractive approach as:  1) it makes use of the stain formation process, by building the network on OD input and the number of network parameters closely mimics the number of physical reference dye vectors used in the staining process  2) Addresses the relatively common issue on how to handle stained slides resultant from different staining protocols.

Pros/Cons:
The paper is  incomplete. What would be interesting follow up questions to be pursued are:

 A) understand the differences in image reconstruction appearance and specific trends/patterns in the classification errors from different methods.

B) How consistent and effective is this method (or other methods) for other HIA applications :
	1) like how does it impact the image appearance in the WS image ( an important question for manual reading)
	2)  in algorithmic detection and classification of other micro and macro structures (such as lymphocytes, blood vessels, macrophages etc, glandular structures and muscular structures) for various tissue type slides.

C) The provided quantitative results and image patches are not adequate enough to conclusively show the improvements of the proposed method over others, in regards to the important challenges in automated interpretation of slides with staining variations mentioned in the introduction section.  The authors need to provide specific esults (may be online) for couple of whole slide images and different tissue types to show that the color normalization method is effective.

Significance:
The work can generate some interesting discussion on A) the need for color standardization in the context of CNN-based classification methods. B) Methods and impact of color normalization on downstream automated H&E analysis for various tissue types and manual reads of digitized slides.

---

### Decision · Program_Chairs · 2019-08-20

Accept